# Carbohydrate sulfotransferase 14 gene deletion induces dermatan sulfate deficiency and affects collagen structure and bowel contraction

Fumiko Ono[1], Yuki Takahashi[1,2], Shin Shimada[3], Shuji Mizumoto[4], Shinji Miyata[5], Yuko Nitahara-Kasahara[6], Shuhei Yamada[4], Takashi Okada[6], Tomoki Kosho[1,2,7,8,9], Takahiro Yoshizawa[3]*

1 Department of Medical Genetics, Shinshu University School of Medicine, Matsumoto, Nagano, Japan, 2 Division of Clinical Sequencing, Shinshu University School of Medicine, Matsumoto, Nagano, Japan, 3 Division of Animal Research, Research Center for Advanced Science and Technology, Shinshu University, Matsumoto, Nagano, Japan, 4 Department of Pathobiochemistry, Faculty of Pharmacy, Meijo University, Nagoya, Japan, 5 Graduate School of Agriculture, Tokyo University of Agriculture and Technology, Tokyo, Japan, 6 Division of Molecular and Medical Genetics, The Institute of Medical Science, The University of Tokyo, Tokyo, Japan, 7 Center for Medical Genetics, Shinshu University Hospital, Matsumoto, Nagano, Japan, 8 Research Center for Advanced Science and Technology, Shinshu University, Matsumoto, Nagano, Japan, 9 BioBank Shinshu, Shinshu University Hospital, Matsumoto, Nagano, Japan

* tyoshizawa@shinshu-u.ac.jp

## Abstract

Dermatan sulfate (DS) is a type of glycosaminoglycan present in the extracellular matrix, and which is related to tissue strength, structure, and healing. Dermatan 4-*O*-sulfotransferase 1 (D4ST1) is an enzyme that catalyzes the transfer of a sulfate group to the *N*-acetylgalactosamine residue of dermatan, resulting in mature DS. Biallelic loss-of-function variants in the carbohydrate sulfotransferase 14 (*CHST14*) gene encoding D4ST1, induce defective DS biosynthesis. DS deficiency causes severe connective tissue fragility and deformities in humans (musculocontractural Ehlers–Danlos Syndrome [mcEDS]) and mice (*Chst14* gene knockout [*Chst14*$^{-/-}$] mice). Many patients with mcEDS experience gastrointestinal symptoms such as constipation, diverticula, diverticulitis, and perforation. However, pathogenesis of these symptoms has not been systematically investigated. Therefore, we sought to determine the effects of DS deficiency on the colon using *Chst14*$^{-/-}$ mice. We found that collagen fibrils were abnormally arranged in the submucosa of the colon. The mice also exhibited accelerated colonic contraction. Unexpectedly, no significant aggravation of dextran sulfate sodium-induced colitis was observed in *Chst14*$^{-/-}$ mice compared with wild-type mice. These findings suggest a physiological role of DS in the colon and may shed light on the potential mechanisms underlying the gastrointestinal symptoms of mcEDS.

**Data availability statement:** All relevant data are within the paper and its Supporting Information files.

**Funding:** JSPS KAKENHI Grant Numbers JP23K07780 (to T.Y.) and 23K06142 (to S.M.). Grant-in-Aid by the Nagano Society for the Promotion of Science (to T.Y.). Grant-in-Aid for Scientific Research (A) from the Japan Society for the Promotion of Science, Grant Number 24H00646, and AMED under Grant Number, JP24bm1523007 (to T.O.). The funders had no role in study design, data collection and analysis, decision to publish, or preparation of the manuscript.

**Competing interests:** Y.T. and T.K. are members of an endowed chair named "Division of Clinical Sequencing, Shinshu University School of Medicine" sponsored by BML Inc. and Life Technologies Japan Ltd. of Thermo Fisher Scientific Inc.

## Introduction

Dermatan sulfate (DS) is present in the extracellular matrix of various tissues such as the skin, tendon, and colon [1]. DS is a type of glycosaminoglycan (GAG), which is found in chondroitin sulfate (CS)/DS long polysaccharide chains composed of repeating disaccharide units of glucuronic acid (GlcA) or iduronic acid (IdoA) and N-acetylgalactosamine (GalNAc). These polysaccharide chains bind to serine residues of core proteins, forming proteoglycans (PGs) [2–4]. CS/DS-PGs interact with diverse molecules including matrix molecules, growth factors, protease inhibitors, cytokines, and chemokines, and are involved in various biological processes such as anticoagulation, infection, healing, and extracellular matrix construction [3]. DS is distinguished from CS by epimerized IdoA at C-5 of GlcA [3]. Dermatan sulfate epimerase (DSE) epimerizes GlcA residues to IdoA residues by C-5 inversion at the polymer level of the substrate, $(-GlcA\beta-1,3-GalNAc\beta-1,4-)_n$. Shortly thereafter, C4 of GalNAc (in the IdoA-GalNAc sequence) is sulfated by dermatan-4-sulfotransferase-1 (D4ST1) [5–7]. This sulfation prevents reverse epimerization from IdoA to GlcA, thereby completing DS biosynthesis [7,8].

DSE and D4ST1 are encoded by *DSE* and carbohydrate sulfotransferase 14 (*CHST14*), respectively. Biallelic loss-of-function variants in *CHST14* and *DSE* result in defective DS biosynthesis, leading to musculocontractural Ehlers–Danlos Syndrome (mcEDS) [9–12]. DS is undetectable in skin fibroblasts or urine samples from patients with mcEDS-*CHST14* [10,13], and significantly reduced in skin fibroblasts or undetectable in urine samples from patients with mcEDS-*DSE* [11,14]. Patients with mcEDS exhibit characteristic clinical manifestations, including multiple malformations and progressive connective tissue fragility-related complications [12,15,16]. An international collaborative study on mcEDS-*CHST14* reported gastrointestinal symptoms as common manifestations, including constipation (85%) and diverticula (35%) [16]. Several patients developed fatal colonic perforations [16–18].

*Chst14* gene knockout (*Chst14*[-/-]) mice and mice generated using CRISPR/Cas9-mediated genomic editing for *Chst14* lack DS [19,20]. Phenotypic features of these mice include postnatal generalized growth disturbance [21], skin fragility with decreased skin tensile strength [21,22], thoracic kyphosis [20], embryonic lethality thought to be associated with placental vascular abnormalities [21,23], and tooth and tail abnormalities [21]. These skin and spine features reflect those observed in patients with mcEDS-*CHST14* [24]. However, the functions of DS and CS/DS-PGs in the intestinal tract have not been investigated.

Therefore, we used *Chst14*[-/-] mice to determine the effects of DS deficiency on the gastrointestinal tract, particularly the colon. We found that *Chst14*[-/-] mice had abnormally arranged collagen fibrils in the submucosa of the colon. They exhibited accelerated intestinal contraction without constipation or diverticula. Unexpectedly, no significant aggravation of dextran sulfate sodium (DSS)-induced colitis was observed in *Chst14*[-/-] mice compared with wild-type (*Chst14*[+/+]) mice. These results suggest a role for DS in the colon and also as the pathological basis of gastrointestinal symptoms in mcEDS.

## Materials and methods

### Animal studies

The mouse strain, B6;129S5-*Chst14*[tm1Lex]/Mmucd (identification number 031629-UCD), was obtained from the Mutant Mouse Regional Resource Center (MMRRC; UC Davis, Sacramento, CA, USA; https://www.mmrrc.org/) [25], an NCRR/ NIH-funded strain repository, and was donated to the MMRRC by Lexicon Genetics Inc and backcrossed on a BALB/ cAJc1(BALB/cA) inbred strain (CLEA Japan Inc., Shizuoka, Japan) [26]. Mice were housed at a constant temperature of 23±3 °C, with a relative humidity of 45%–70% and a 12-hour light/dark cycle. Animals had free access to tap water and standard mouse chow (Funabashi Farm, Chiba, Japan). In this study, male mice were used to ensure consistency in the experimental conditions. This approach minimizes gender-related variations in body weight and colon length, allowing for a more direct evaluation of the effects of the experimental conditions. In the present study, no invasive treatment was performed on the live mice. Animal welfare was closely monitored throughout the experiments, and no severe pain was observed in the mice. The mice were euthanized by cervical dislocation before sample collection. All experimental procedures were performed in accordance with the Regulations for Animal Experimentation of Shinshu University. The animal protocol was reviewed by the Committee for Animal Experiments of Shinshu University based on national regulations and guidelines, and approved by the president of Shinshu University (Approval number: 022010).

### Genotyping

The PCR method was reported in our previous studies [23,26]. Ear specimens were collected from 3-week-old mice and DNA was extracted using Mighty Prep reagent (Takara Bio Inc., Shiga, Japan). Primer sequences for wild-type genotyping designed to exon 1 of the *Chst14* gene were: 5′-GGACCACCGCAGTGACTTG-3′ and 5′-ACAGGCATCCAATGCT CATTC-3′. Primer sequences for the neomycin resistance gene in knockout PCR were: 5′-TGGCTCTCCTCAAGCG TATT-3′ and 5′-GTTTTCCCAGTCACGACGTT-3′. PCR was performed using TaKaRa Taq™ HS Perfect Mix (Takara Bio Inc.). The PCR conditions were 94 °C for 1 minute followed by 30 cycles at 94 °C for 5 seconds and then 65 °C for 15 seconds. PCR products were visualized using agarose gel electrophoresis.

### Analysis of food intake, body weight, and fecal volume

To avoid coprophagy, 7-week-old male mice were housed individually in wire mesh cages and allowed to acclimatize for 4 days. Food intake, body weight, and fecal volume were measured at fixed times. The defecation volume was measured using a precision balance (Shimadzu Corporation, Kyoto, Japan).

### Bead expulsion time

Ten to thirteen-week-old male mice were anesthetized with 1.5% isoflurane (Pfizer Inc., Manhattan, NY, USA) using an inhalation anesthesia apparatus (MK-AT210D; Muromachi Kikai Co., Ltd., Kyoto, Japan). A 5-mm bead was inserted into the anus and the time to expulsion was measured.

### Intestinal transit time

Ten to eleven-week-old male mice were orally administered 200 µL of 6% carmine red and 0.5% methylcellulose (Fujifilm Wako Pure Chemical Co., Osaka, Japan) in phosphate-buffered saline (PBS). The cages were inspected every 10 minutes after oral administration, and time of appearance of the first red fecal pellet was recorded [27].

### Dextran sulfate sodium-induced colitis model

Colon injury was induced by 2.0% (weight/volume) DSS (molecular weight 36,000–50,000 g/mol; MP Biomedicals, Santa Ana, CA, USA) in drinking water [28]. Eight-week-old male *Chst14*[+/+] and *Chst14*[-/-] mice were randomly assigned to three

groups: Control group, 2.0% DSS (day 8) group, and 2.0% DSS (day 15) group. The 2.0% DSS groups were given 2.0% DSS solution in drinking water *ad libitum* for 7 days, while the Control group received sterilized water under the same conditions. Dissections were performed on day 8 for the Control group and 2.0% DSS (day 8) group, and on day 15 for the 2.0% DSS (day 15) group. Body weight, fecal characteristics, and water intake were monitored daily at 17:00±2 hours. The disease activity index (DAI) score, which is a composite measure of weight loss, stool bleeding, and stool consistency, was determined as previously reported [29–31]. Body weight loss: 0 (no loss), 1 (1%–5% loss), 2 (5%–10% loss), 3 (10%–20% loss), and 4 (> 20% loss). Stool consistency: 0 (normal), 2 (loose stool), and 4 (diarrhea, gross bleeding). Hematoxylin and eosin (H&E) scoring was used for histological determination of colon injury [32,33]. Inflammation: 0 (no inflammation), 1 (mild), 2 (moderate), and 3 (severe). Inflammatory cell infiltration: 0 (none), 1 (mucosal layer), 2 (submucosal layer), and 3 (all muscle layers). Injury to crypts: 0 (1/3), 1 (2/3), 3 (surface only), and 4 (loss of mucosal layer); and range: 1 (1%–25%), 2 (26%–50%), 3 (51%–75%), and 4 (76%–100%).

## Sample collection

Mice were euthanized and their colons were collected. Photographs of the tissue were obtained. Colons were then washed in PBS and fixed with 4% paraformaldehyde, or stored at −80 °C. For experiments with nifedipine (Fujifilm Wako Pure Chemical Co.), colons were immersed in PBS containing 4 μM nifedipine for 10 minutes.

## Strength and extensibility test

Colons of male mice aged 8–10 weeks were cut along the longitudinal axis into cylindrical strips 3 mm wide. A 10-g weight was loaded and the length of each extended colon sample was measured. Colon strength (force required for rupture) was measured using a dynamometer (Muromachi Kikai Co., Ltd.).

## Pathological analysis

Samples were fixed in 4% paraformaldehyde for 48 hours at 4 °C. After dehydration in an ethanol and xylol series, whole colon samples were embedded in paraffin and cut into 5 μm thick sections. Sections were deparaffinized and then subjected to hydrophilic treatment by stepwise ethanol washes. This was followed by deionized water and then used for H&E, Sirius red, and Alcian blue staining. Thirty random locations were selected from each sample stained with H&E to measure the mucosal thickness, mucosal muscle, submucosa, and muscle layers using ImageJ software [34]. H&E staining and H&E scores were used to determine the extent of tissue damage. To observe collagen components in the colon, Sirius red staining was performed using Weigert's iron hematoxylin and Von Gieson's solution, according to the manufacturer's instructions (Muto Pure Chemicals Co., Ltd., Tokyo, Japan). Alcian blue staining using 3% acetic acid (Fujifilm Wako Pure Chemical Corp.,), Alcian blue solution, and Kernechtrot solution (Muto Pure Chemicals Co., Ltd.) was performed to stain mucins and GAGs for assessment of damage to the glandular ducts of the mucosal layer.

## Immunofluorostaining of lymphocyte antigen-6 family member G (Ly-6G)

Immunofluorostaining of Ly-6G was performed on 6 μm thick paraffin sections. After deparaffinization and hydrophilic treatment, antigen retrieval was performed by autoclaving with HISTOFINE Depara Antigen Activation Solution, pH 6 (Nichirei, Tokyo, Japan). After blocking with 2% bovine serum albumin, sections were incubated overnight at 4 °C with a 1:50 dilution of Alexa Fluor 488-conjugated Ly-6G antibody (Thermo Fisher Scientific, Waltham, MA, USA). Nuclei were then stained with a 1:1000 dilution of 4′,6-diamidino-2-phenylindole (DAPI) (Invitrogen, Waltham, MA, USA). Stained sections were observed using a fluorescence microscope (Olympus, Tokyo, Japan). Ly-6G-positive cells in all sections were counted three times and averaged.

## Transmission electron microscopy (TEM)

TEM was used to observe the condition of collagen fibrils. Colon tissue was sectioned using a scalpel and fixed with 2.5% glutaraldehyde and 4% osmium tetroxide. It was then embedded in epoxy resin, cut into ultrathin sections, and stained with uranyl acetate and lead citrate. Carbon shadowing was applied, and sections were observed by TEM (JEM-1400; JEOL, Tokyo, Japan). Three samples from each group of $Chst14^{+/+}$ and $Chst14^{-/-}$ mice were used to measure collagen fibril diameters using ImageJ software [34].

## Scanning electron microscopy (SEM)

Fresh colon samples were fixed with 2.5% glutaraldehyde for 2 days at 4 °C. They were then treated with 8% NaOH at room temperature for 7–10 days, followed by washing with distilled water for 3–7 days. Samples were treated with 1% tannic acid for 2 hours and washed with distilled water. Next, samples were postfixed in 1.0% osmium tetroxide in 0.1 M phosphate buffer for 1 hour at room temperature and washed with distilled water. Samples were then dried using the t-butyl alcohol freeze-drying method, mounted on metal stubs, coated with platinum, and observed by SEM (JSM-7600F; JEOL) with an acceleration voltage of 5 kV.

## Western blot analysis

Colon tissues from 10-week-old male mice were homogenized in PBS containing 1% Triton X-100 and a protease inhibitor cocktail (Merck, Darmstadt, Germany), followed by incubation on ice for 30 minutes. The homogenates were centrifuged at 20,000 × g for 30 minutes at 4 °C, and protein concentrations in the supernatants (colon lysates) were measured using a BCA assay kit (Thermo Fisher Scientific, Waltham, MA, USA). Colon lysates (100 μg protein) were treated with 5 milliunits of chondroitinase ABC (Merck, Darmstadt, Germany) or chondroitinase B (R&D Systems, Minneapolis, MN, USA) at 37 °C for 4 hours. Both digested and undigested lysates were denatured with 2% sodium dodecyl sulfate and 5% mercaptoethanol at 60 °C for 20 minutes. Proteins were separated by 7.5% polyacrylamide gel electrophoresis and transferred to polyvinylidene difluoride membranes. The membranes were blocked with 2% skim milk in PBS containing 0.1% Tween-20 and then incubated overnight at 4 °C with primary antibodies: anti-mouse decorin (DCN) antibody (goat IgG, AF1060, 1:2000, R&D Systems, Minneapolis, MN, USA) or anti-glyceraldehyde-3-phosphate dehydrogenase (GAPDH) antibody (mouse IgG, clone 5A12, 1:10,000, Fujifilm Wako Pure Chemical Co., Osaka, Japan). After washing, the membranes were incubated with appropriate horseradish peroxidase-conjugated secondary antibodies (1:4000) for 1 hour at room temperature. Protein signals were visualized using Immobilon Western Chemiluminescent HRP Substrate (Merck, Darmstadt, Germany) and imaged using a LuminoGraph I imaging system (ATTO, Tokyo, Japan).

## Quantitative analysis of DS and CS disaccharides

Colons were washed, homogenized, and sonicated. Extraction and purification of GAG fractions were performed as described previously [35]. Samples were treated individually with chondroitinase B (EC4.2.2.19) (R&D Systems, Minneapolis, MN, USA) or a mixture of chondroitinase ABC and AC-II (EC4.2.2.5) (Seikagaku Corp., Tokyo, Japan), respectively. The disaccharide composition of DS or CS/DS moieties on CS/DS chains was analyzed. Digests were labeled with the fluorophore, 2-aminobenzaimide (2AB), and samples were analyzed for each disaccharide by anion-exchange high-performance liquid chromatography (HPLC) on a PA-G column (YMC Co., Kyoto, Japan), as described previously [35]. Disaccharide standards were also labeled with 2AB and analyzed by HPLC. The standards used were: ΔHexUA-GalNAc, ΔHexUA-GalNAc(6S), ΔHexUA-GalNAc(4S), ΔHexUA(2S)-GalNAc(6S), ΔHexUA(2S)-GalNAc(4S), ΔHexUA-GalNAc(4S,6S), and ΔHexUA(2S)-GalNAc(4S,6S). Unsaturated DS and CS/DS disaccharides observed in digests were identified by comparison with elution positions of 2AB-labeled disaccharide standards and quantitated based on peak area relative to standard unsaturated disaccharides.

### Real-time quantitative reverse transcription PCR

Frozen colons were homogenized in TRI reagent (Molecular Research Center, Cincinnati, OH, USA) using a Bio-Gen PRO200 homogenizer (PRO Scientific, Oxford, CT, USA) to extract total RNA. Homogenates were treated with DNase and RNA then purified with a RNA Clean and Concentrator Kit (Zymo Research, Irvine, CA, USA). RNA was subjected to reverse transcription to synthesize cDNA using a High-Capacity cDNA Reverse Transcription Kit (Applied Biosystems, Waltham, MA, USA). Quantitative PCR was performed using a QuantStudio 3 Real-Time PCR system (Applied Biosystems) with Thunderbird Next SYBR qPCR Mix (Toyobo, Osaka, Japan). Values were normalized to 18S ribosomal RNA levels. The primer sequences are listed in S1 Table.

### Statistical analysis

Data are reported as mean±standard error of the mean (SEM). Statistical comparisons between groups were performed using GraphPad Prism software Ver 10.2.3 (GraphPad Software, San Diego, CA, USA). Differences between two groups were assessed using unpaired two-tailed Student's *t*-tests. Data from groups of three or more were analyzed using one-way analysis of variance (ANOVA) followed by the Tukey–Kramer post hoc test for multiple comparisons. In all analyses, *P*-values<0.05 were considered statistically significant.

## Results

### DS is absent from the colon of *Chst14-/-* mice

The DS or CS/DS moiety on CS/DS chains from colon samples were digested into disaccharides with chondroitinase ABC and AC-II or chondroitinase B, respectively. HPLC profiles were obtained (Fig 1A–D). Regarding CS/DS disaccharide levels, ΔHexUA-GalNAc, ΔHexUA(2S)-GalNAc(6S), and ΔHexUA(2S)-GalNAc(4S,6S) were not detected (ΔHexUA[2S, 4S, and 6S] represent 4,5-unsaturated hexuronic acid, 2-*O*-, 4-*O*-, and 6-*O*-sulfate, respectively). DS disaccharides were only detected in *Chst14*[+/+] mice and not *Chst14*[-/-] mice. ΔHexUA-GalNAc(6S) and ΔHexUA-GalNAc(4S,6S) are component solely of CS disaccharides. Only ΔHexUA-GalNAc(6S) was significantly increased in *Chst14*[-/-] mice. CS disaccharide levels of ΔHexUA-GalNAc(4S), which is component of CS/DS disaccharides, were significantly higher in *Chst14*[-/-] mice than that of *Chst14*[+/+] mice. Mean but while, CS/DS disaccharide levels were significantly lower in *Chst14*[-/-] mice than that of *Chst14*[+/+] mice. ΔHexUA(2S)-GalNAc(4S) was mostly composed of DS disaccharides. Total CS/DS disaccharides were significantly lower in *Chst14*[-/-] mice than *Chst14*[+/+] mice (Fig 1E). Western blot was performed to analyze GAG modifications on DCN, a major CS/DS-PG, in the colon of *Chst14*[+/+] and *Chst14*[-/-] mice. In undigested colon lysates, DCN from both *Chst14*[+/+] and *Chst14*[-/-] mice showed high molecular weight signals indicative of GAG modification. When digested with chondroitinase ABC, which degrades both DS and CS, and chondroitinase B, which specifically degrades DS, DCN from *Chst14*[+/+] mice shifted similarly to a lower molecular weight corresponding to the position of the core protein. This finding suggests that DCN in the colon of *Chst14*[+/+] mice is predominantly modified by DS. In contrast, DCN from the colon of *Chst14*[-/-] mice was resistant to chondroitinase B but was completely digested by chondroitinase ABC. These results indicate that in the absence of *Chst14*, DS on DCN in the colon is significantly reduced and replaced with CS (Fig 1F). *Chst14* gene expression was not detected in *Chst14*[-/-] mice. Expression of other genes related to CS/DS synthesis (*Chst3*, *Chst11*, *Chst12*, *Chst15*, uronyl 2-sulfotransferase [*Ust*], and *Dse*) were not significantly different between the two groups of mice (Fig 1G).

### *Chst14*[-/-] mice have shorter colons with increased contractility

The colons of 10-week-old *Chst14*[-/-] mice were shorter compared with *Chst14*[+/+] mice. However, no anatomical abnormalities or diverticula were observed (Fig 2A, C). *Chst14*[-/-] mice weighed significantly less than *Chst14*[+/+] mice (Fig 2B). To account for weight-related biases, we compared the colon length of 10-week-old *Chst14*[-/-] mice with 6-week-old *Chst14*[+/+]

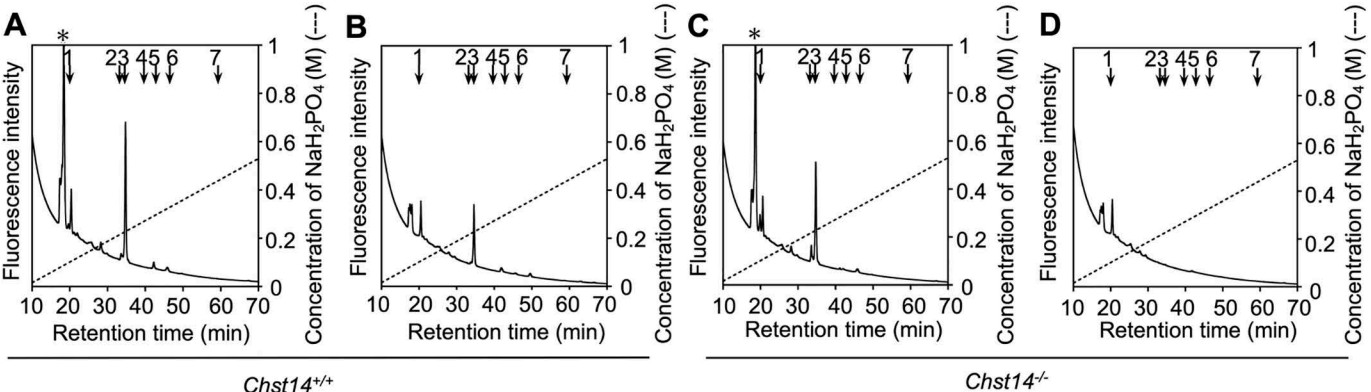

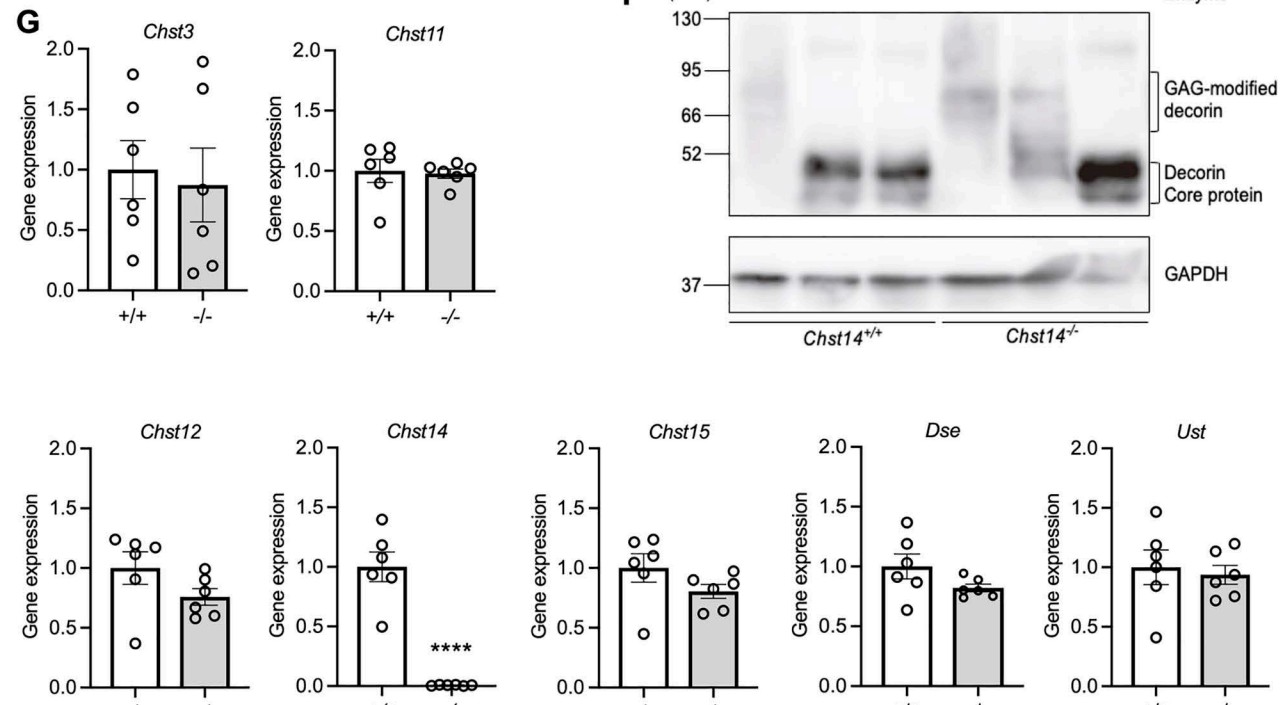

**E**

| Disaccharides composition | CS/DS | | CS | | DS | |
|---|---|---|---|---|---|---|
| | pmol/mg protein | | | | | |
| | $Chst14^{+/+}$ | $Chst14^{-/-}$ | $Chst14^{+/+}$ | $Chst14^{-/-}$ | $Chst14^{+/+}$ | $Chst14^{-/-}$ |
| ΔHexUA-GalNAc | N.D. | N.D. | N.D. | N.D. | N.D. | N.D. |
| ΔHexUA-GalNAc(6S) | 35.1±2.4 | 87.0±6.9[*] | 35.1±2.4 | 87.0±7.0[*] | N.D. | N.D. |
| ΔHexUA-GalNAc(4S) | 873.5±30.6 | 666.3±26.6[**] | 414.5±11.0 | 666.3±26.6[***] | 459.0±23.8 | N.D.[****] |
| ΔHexUA(2S)-GalNAc(6S) | N.D. | N.D. | N.D. | N.D. | N.D. | N.D. |
| ΔHexUA(2S)-GalNAc(4S) | 39.1±3.2 | N.D.[***] | 0.3±3.8 | N.D. | 38.7±3.7 | N.D.[***] |
| ΔHexUA-GalNAc(4S,6S) | 44.9±7.4 | 42.1±1.6 | 44.9±7.4 | 42.1±1.6 | N.D. | N.D. |
| ΔHexUA(2S)-GalNAc(4S,6S) | N.D. | N.D. | N.D. | N.D. | N.D. | N.D. |
| Total | 992.6±38.1 | 795.4±34.6[***] | 494.9±13.7 | 795.4±34.5[**] | 497.7±25.9 | N.D.[****] |

**Fig 1. HPLC profiles of DS and CS/DS moieties and expression levels of CS/DS synthesis genes.** (A) HPLC profiles of CS/DS digests (A, C) and DS (B, D) moieties in CS/DS chains prepared from colons of $Chst14^{+/+}$ mice (A, B) and $Chst14^{-/-}$ mice (C, D). Colons were digested into disaccharides for analysis of CS/DS or DS moieties using chondroitinase ABC and AC-II (A, C) or chondroitinase B (B, D), respectively. Each digest was labeled with 2AB,

and labeled CS/DS disaccharides were separated by anion-exchange HPLC on amine-bonded silica PA-G columns using a linear gradient of $NaH_2PO_4$, as indicated by the dashed line. The amounts of disaccharides in each sample were calculated based on peak area relative to standard unsaturated disaccharides. Elution positions of standard 2AB-labeled CS/DS disaccharides are indicated by numbered arrows: 1, ΔHexUA-GalNAc; 2, Δ HexUA-GalNAc(6S); 3, ΔHexUA-GalNAc(4S); 4, ΔHexUA(2S)-GalNAc(6S); 5, ΔHexUA(2S)-GalNAc(4S); 6, ΔHexUA-GalNAc(4S,6S); and 7, ΔHexUA(2S)-GalNAc(4S,6S), where ΔHexUA, 2S, 4S, and 6S represent 4,5-unsaturated hexuronic acid, 2-*O*-sulfate, 4-*O*-sulfate, and 6-*O*-sulfate, respectively. An asterisk indicates ΔHexUA-*N*-acetylglucosamine derived from hyaluronan. (E) Disaccharide composition of CS/DS, CS, and DS in *Chst14*-/- and *Chst14*+/+ mice. The disaccharide contents of CS/DS and DS were calculated from the peak areas of the HPLC profiles the CS disaccharide content was obtained by subtracting DS from CS/DS. Each value is the mean±SEM. *Chst14*+/+ and *Chst14*-/- mice (n=3 per group). *$P<0.05$, **$P<0.01$, ***$P<0.001$, ****$P<0.0001$, *t*-test. N.D.: not detected (< 0.1 pmol/mg protein). (F) Upper panel: Western blot analysis of DCN in colon lysates from *Chst14*+/+ and *Chst14*-/- mice after digestion with chondroitinase B or chondroitinase ABC. Lower panel: Western blot analysis of GAPDH used as a loading control. (G) Relative mRNA levels (mean±SEM) of *Chst3*, *Chst11*, *Chst12*, *Chst14*, *Chst15*, *Ust*, and *Dse* in colon samples from *Chst14*+/+ and *Chst14*-/- mice (n=3 per group). ****$P<0.0001$, *t*-test. DS, dermatan sulfate; CS, chondroitin sulfate; HPLC, high-performance liquid chromatography; *Chst3*, carbohydrate sulfotransferase 3; *Chst11*, carbohydrate sulfotransferase 11; *Chst12*, carbohydrate sulfotransferase 12; *Chst14*, carbohydrate sulfotransferase 14; *Chst15*, carbohydrate sulfotransferase 15; *Ust*, uronyl-2-sulfotransferase; *Dse*, dermatan sulfate epimerase; *Chst14*+/+ mice, +/+; *Chst14*-/- mice, -/-.

mice matched for body weight (Fig 2D). Even after this adjustment, the colons of *Chst14*-/- mice were still significantly shorter (Fig 2E). Nifedipine, a calcium channel blocker, was used to investigate the effect of intestinal contraction on colon shortening. Nifedipine suppressed intestinal contractions and eliminated the difference in colon length between *Chst14*+/+ and *Chst14*-/- mice (Fig 2A, C).

### Physiological colon function does not differ between *Chst14*+/+ and *Chst14*-/- mice

There were no differences in the amount of daily defecation or food consumption between *Chst14*+/+ and *Chst14*-/- mice (Fig 2F, G). There was also no difference in gastrointestinal transit time (Fig 2H). Additionally, there was no difference in the rectoanal reflex, determined by inserting a bead into the rectum and recording the time of expulsion (Fig 2I). The rate of colon extension and load required for colon rupture showed no significant differences between *Chst14*+/+ and *Chst14*-/- mice (Fig 2J, K).

### Collagen fibril deformation is induced by DS deficiency in the colon of *Chst14*-/- mice

H&E staining revealed no histological differences in distal colon tissue between *Chst14*+/+ and *Chst14*-/- mice (Fig 3A). There were no differences in the thickness of each tissue layer (Fig 3B). Sirius red is used to stain collagen fibers, which are mainly localized in the submucosa (Fig 3A). These submucosal collagen fibrils were observed using TEM, and found to be sparse and disorganized in *Chst14*-/- mice (Fig 3A). Using SEM, *Chst14*-/- mice had a lower density of collagen fibrils than *Chst14*+/+ mice (Fig 3A). Collagen fibrils had significantly smaller diameters in *Chst14*-/- mice than *Chst14*+/+ mice (Fig 3C). There were no differences in gene expression of collagen 1 alpha-1 (*Col1a1*) and collagen 3 alpha-1 (*Col3a1*), components of collagen in the colon (Fig 3D). There were also no differences in gene expression of *Dcn*, biglycan (*Bgn*), and versican (*Vcan*), which are core proteins of CS/DS-PGs (Fig 3D).

### *Chst14*-/- mice do not have more severe DSS-induced colitis compared with *Chst14*+/+ mice

The DSS-induced colitis model was used to investigate the inflammatory response provoked by intestinal mucosal damage. No difference was observed in the amount of DSS-containing water consumed by *Chst14*-/- and *Chst14*+/+ mice (Fig 4A). The DSS administration model is known to induce weight loss, loose stools, diarrhea, and bloody stools [36]. There were no significant differences between *Chst14*-/- and *Chst14*+/+ mice in DAI scores and body weight changes (Fig 4B, C). Weight loss peaked on day 10 (2 days after termination of DSS administration) and then recovered in both groups (Fig 4C). On day 8 of DSS administration, significant colon shortening was observed in *Chst14*+/+ mice compared with the control group. This shortening was not observed in *Chst14*-/- mice (Fig 4D, E). H&E staining on day 8 of DSS administration showed destruction of glandular structures, ulceration of the mucosal layer, marked edema of the submucosal layer, and

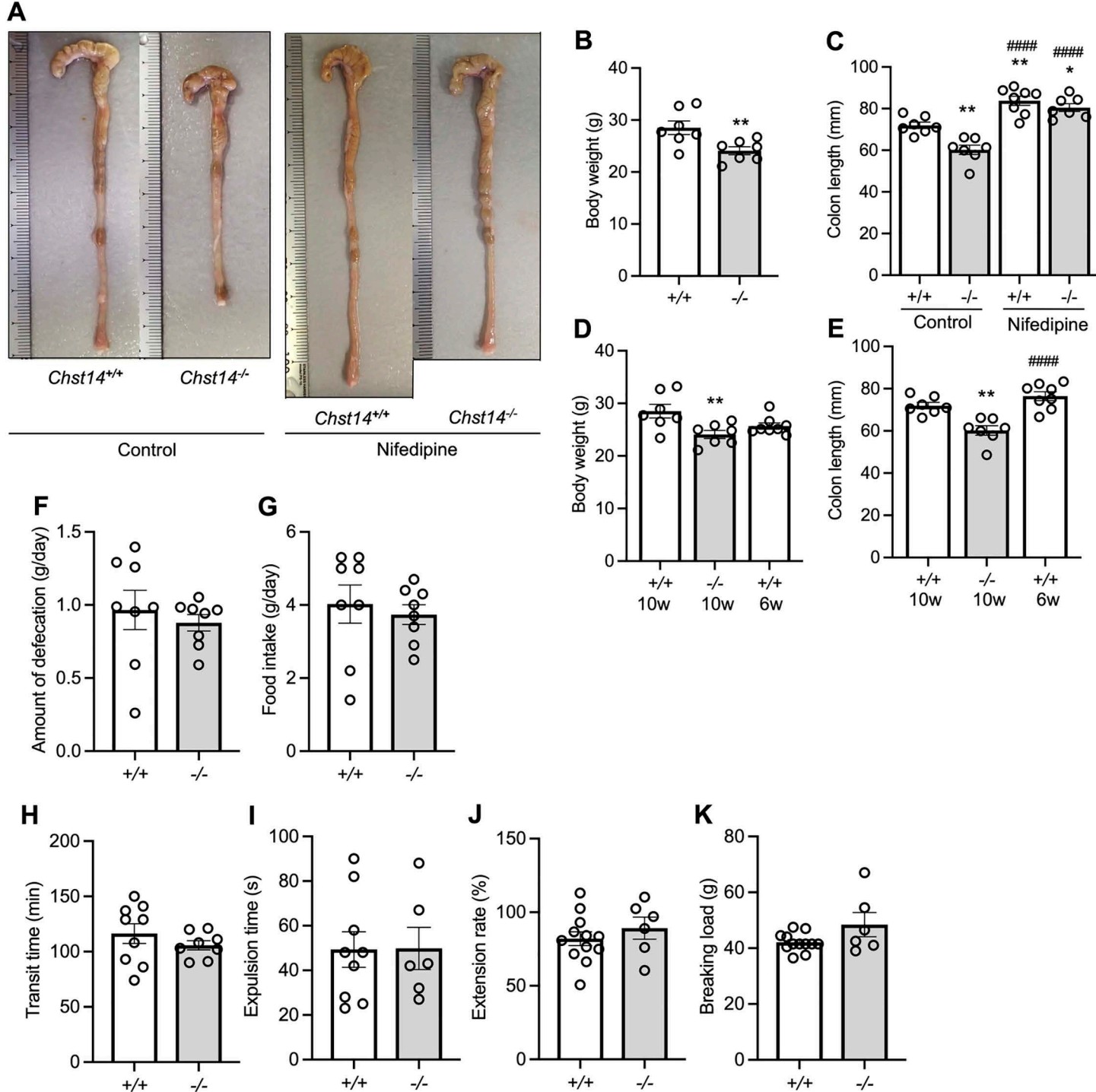

**Fig 2. Appearance and physiological function of the colon.** (A) Typical appearance of control colon and colon treated with nifedipine. (B) Body weight (mean±SEM). Control 10-week-old *Chst14+/+* and *Chst14-/-* mice (n=7 per group). **P<0.01, *t*-test. (C) Comparison of colon length between control and nifedipine groups (mean±SEM). Control *Chst14+/+* mice, n=7; and *Chst14-/-* mice, n=7; and Nifedipine *Chst14+/+* mice, n=8; and *Chst14-/-* mice, n=7. Data were analyzed using one-way ANOVA followed by Tukey–Kramer post hoc test. *P<0.05, **P<0.01 compared with *Chst14+/+* control group. ####P<0.0001 compared with control *Chst14-/-* group. (D) Body weight comparison with 6-week-old *Chst14+/+* mice (mean±SEM). 10-week-old *Chst14+/+* and *Chst14-/-* mice, n=7 per group; and 6-week-old *Chst14+/+* mice, n=8. Data were analyzed using one-way ANOVA followed by Tukey–Kramer post hoc test. **P<0.01 compared with 10-week-old *Chst14+/+* group. (E) Comparison of colon length in weight-matched groups. Colon length of 10-week-old

*Chst14*+/+ and *Chst14*-/- mice, n = 7 per group; and 6-week-old *Chst14*+/+ mice, n = 8. Mean ± SEM. Data were analyzed using one-way ANOVA followed by Tukey–Kramer post hoc test. \*\**P* < 0.01 compared with 10-week-old *Chst14*+/+ group. ####*P* < 0.0001 compared with 10-week-old *Chst14*-/- group. (F) Amount of defecation per day. *Chst14*+/+ and *Chst14*-/- mice, n = 8 per group. (G) Daily food intake. *Chst14*+/+ and *Chst14*-/- mice, n = 8 per group. (H) Gastrointestinal transit time. *Chst14*+/+ mice, n = 9; and *Chst14*-/- mice, n = 8. (I) Bead ejection time. *Chst14*+/+ mice, n = 9; and *Chst14*-/- mice, n = 6. (J) Rate of colon extension when loaded with a 10 g weight. *Chst14*+/+ mice, n = 12; and *Chst14*-/- mice, n = 6. (K) Load required for colon rupture. *Chst14* +/+ mice, n = 12; and *Chst14*-/- mice, n = 6. (F–K) Analyzed using *t*-test. Mean ± SEM. *Chst14*+/+ mice, +/+; *Chst14*-/- mice, -/-.

infiltration of inflammatory cells throughout all layers in both *Chst14*-/- and *Chst14*+/+ mice (Fig 4G). H&E staining on day 15 after the start of DSS administration (7 days after termination of DSS administration) showed improvements in edema in both groups (Fig 4G). However, *Chst14*+/+ mice showed more persistent inflammatory cell infiltration and destruction of the mucosal layer than *Chst14*-/- mice (Fig 4G). Based on H&E scores, there was no significant difference in colon tissue injury between *Chst14*-/- and *Chst14*+/+ mice at day 8. In contrast, *Chst14*-/- mice exhibited a significantly lower H&E score compared with *Chst14*+/+ mice on day 15 (Fig 4F). Alcian blue staining, which detects mucins and GAGs present in the intestinal mucosal glands, identified destroyed glandular structures and decreased mucin on day 8 in both *Chst14*-/- and *Chst14*+/+ mice (Fig 4G). By day 15, *Chst14*-/- mice showed greater amounts of mucin and/or GAGs compared with *Chst14*+/+ mice (Fig 4G). Sirius red staining was diffuse in the disrupted mucosal layer of both *Chst14*-/- and *Chst14*+/+ mice on day 8 of DSS administration (Fig 4G). On day 15, *Chst14*+/+ mice exhibited stronger Sirius red staining in the submucosa, indicating increased collagen hyperplasia compared with *Chst14*-/- mice (Fig 4G). TEM showed disorganized collagen fibril assembly in the submucosa of both *Chst14*-/- and *Chst14*+/+ mice on day 8 of DSS administration (Fig 4G). On day 15, collagen fibrils appeared to reassemble in *Chst14*+/+ mice (Fig 4G).

## Inflammatory response in DSS-induced colitis in DS deficiency mice

Immunostaining for Ly-6G was used to examine neutrophil infiltration. On day 8, the number of Ly-6G-positive cells was similar between *Chst14*-/- and *Chst14*+/+ mice (Fig 5A, B). On day 15, *Chst14*-/- mice tended to have fewer neutrophils compared with *Chst14*+/+ mice (Fig 5A, B). Gene expression of transforming growth factor beta (*Tgfb*) and tumor necrosis factor-alpha (*Tnfa*) were significantly increased on day 8 in both *Chst14*-/- and *Chst14*+/+ mice (Fig 5C). After discontinuing DSS administration (day 15), gene expression of *Tgfb* was decreased in *Chst14*-/- mice only, and continued to rise in *Chst14*+/+ mice. Gene expression of *Tnfa* decreased in both *Chst14*-/- and *Chst14*+/+ mice on day 15 (Fig 5C). Notably, gene expression of interleukin 1 beta (*Il1b*) increased significantly in *Chst14*+/+ mice only on day 8 (Fig 5C).

## Discussion

In this study, we sought to characterize the colonic phenotype of *Chst14*-/- mice to investigate the effects of DS deficiency. The colons of *Chst14*-/- mice exhibited a loss of DS disaccharides, shortening due to increased contractility, smaller collagen fibril diameters, and abnormal submucosal collagen fibril networks. There were no significant differences in physiological indicators and colon strength or extensibility. DSS colitis in *Chst14*-/- mice showed no aggravation compared with *Chst14*+/+ mice. These findings may enhance our understanding of DS function in the colon.

Neither DS nor *Chst14* gene expression were detected in the colons of *Chst14*-/- mice. On the other hand, CS synthesis was elevated in the colon compared with *Chst14*+/+ mice. The compensatory increase in CS was thought to be due to accumulation of dermatan caused by loss of D4ST1 and promotion of chondroitin synthesis with reverse isomerization by DSE [37,38]. Aside from *Chst14*, the gene expression of other enzymes associated with CS/DS synthesis was similar in the colons of *Chst14*-/- and *Chst14*+/+ mice. These results indicate that the compensatory increase in CS in the colon of *Chst14*-/- mice occurs within the normal range of gene expression of relevant CS/DS synthetic enzymes.

H&E staining of colon tissue from *Chst14*-/- mice showed no obvious pathological differences, as previously reported in humans [17]. However, electron microscopy detected poorly aggregated collagen fibrils with reduced density and smaller

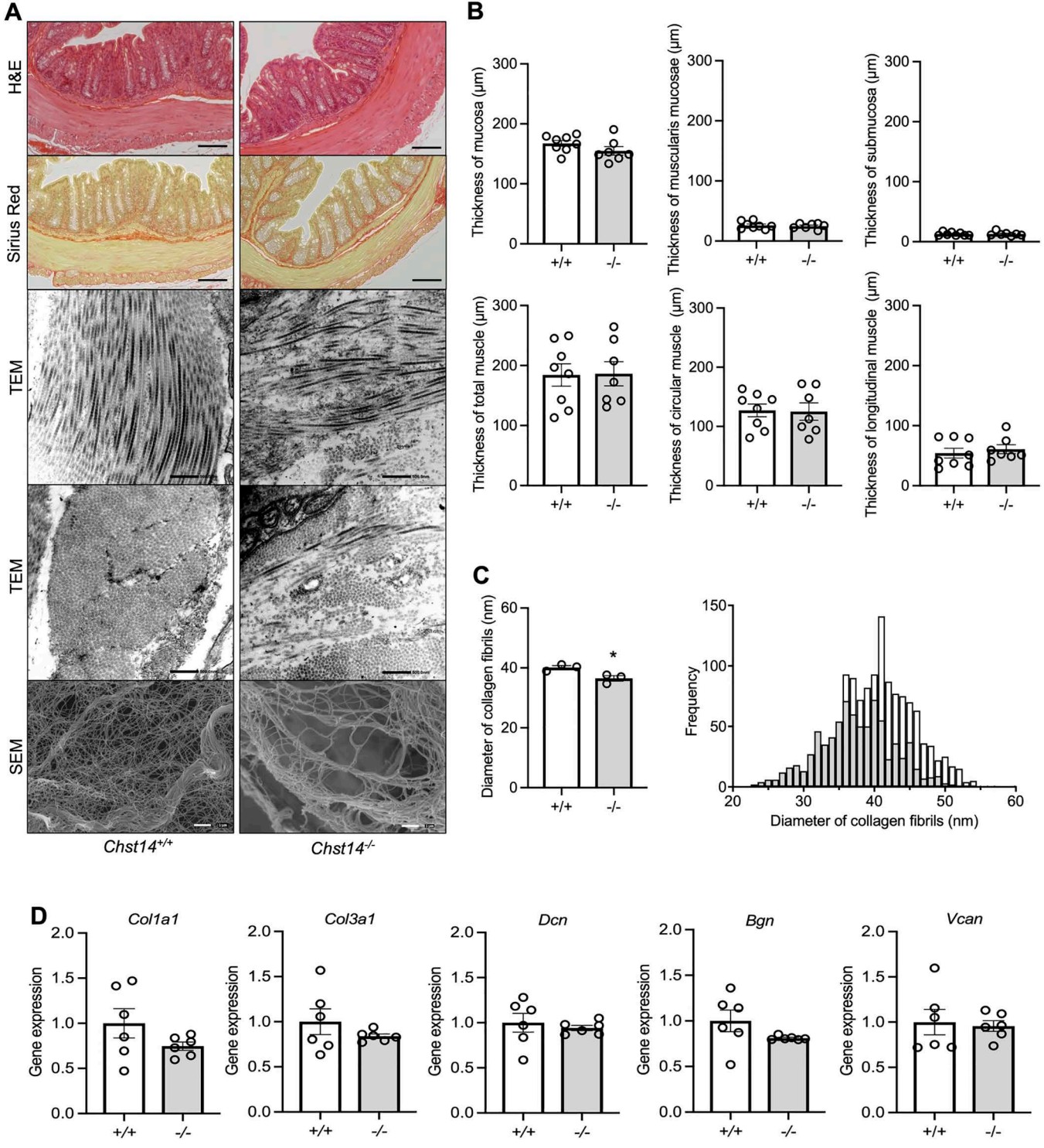

**Fig 3. Pathological findings in *Chst14⁻/⁻* mice and changes in collagen fibrils due to DS deficiency.** (A) H&E- and Sirius red-stained colon sections from *Chst14⁺/⁺* and *Chst14⁻/⁻* mice. Scale bar: 200 μm. Longitudinal and transverse sections of collagen fibrils in the submucosa of the colon observed by TEM. Scale bar: 500 nm. Collagen fibrils in the submucosa of the colon observed by SEM. Scale bar: 1 μm. (B) Colon layer thickness. *Chst14⁺/⁺* and *Chst14⁻/⁻* mice, n = 8 per group. (C) Comparison of collagen diameter (mean ± SEM). *Chst14⁺/⁺* and *Chst14⁻/⁻* mice, n = 3 per group. *P < 0.05, *t*-test. Histogram of collagen fibril diameter. White bars: collagen fibrils in *Chst14⁺/⁺* mice, n = 1306; and gray bars: collagen fibrils in *Chst14⁻/⁻* mice, n = 806. (D) Relative mRNA levels of *Col1a1*, *Col3a1*, *Dcn*, *Bgn*, and *Vcan* in the colon (mean ± SEM) analyzed by *t*-test. *Chst14⁺/⁺* and *Chst14⁻/⁻* mice, n = 6 per group. *Chst14⁺/⁺* mice, +/+; *Chst14⁻/⁻* mice, -/-.

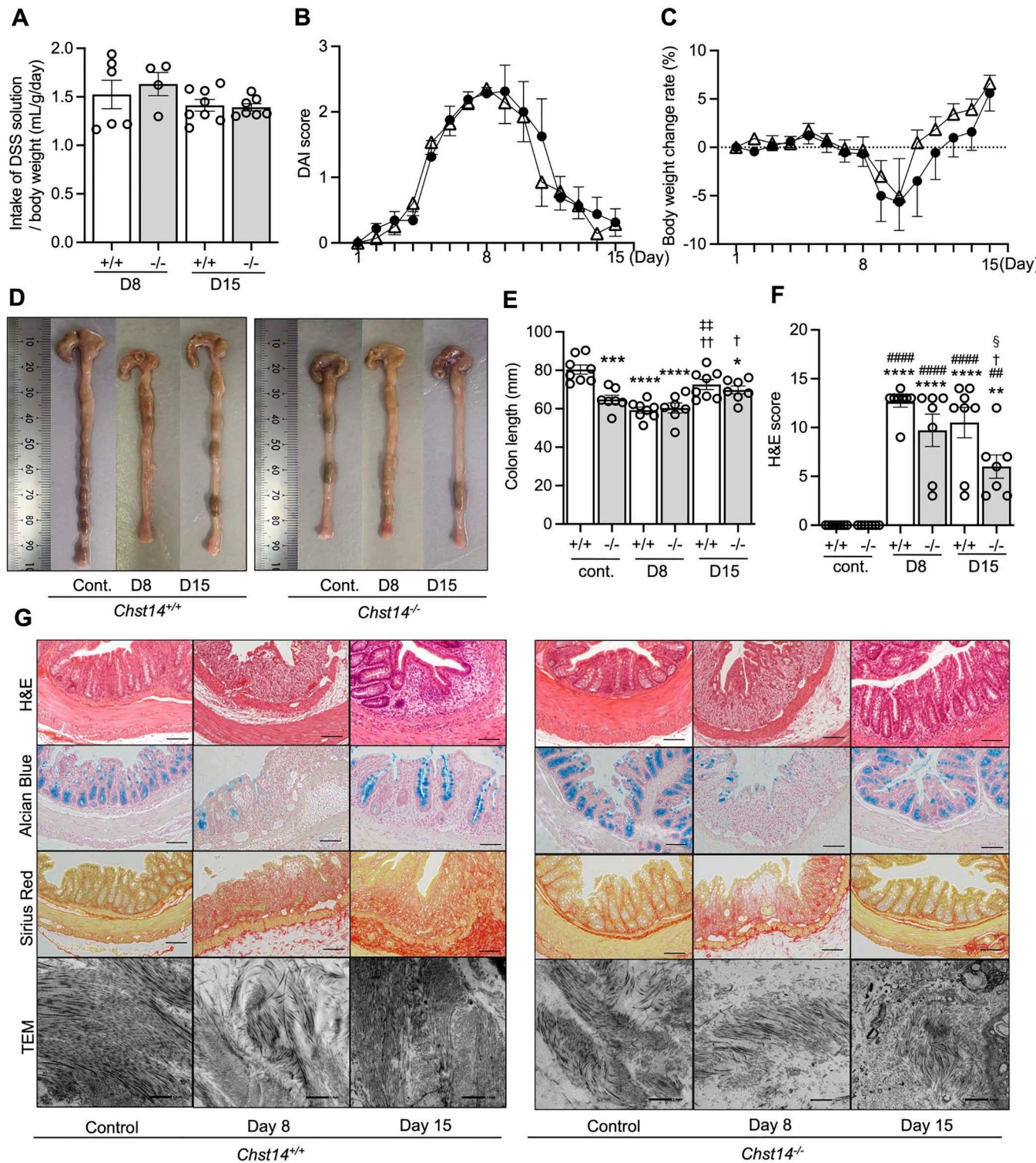

**Fig 4. Experimental course and inflammatory response in the DSS colitis model.** (A) Intake of DSS solution (mean ± SEM). Volume of DSS solution was adjusted using a specific gravity of 1.016. (B) Change in DAI scores (mean ± SEM). (C) Percentage changes in body weight (mean ± SEM). (B, C) Black circle: *Chst14*⁺/⁺ mice; and white triangle: *Chst14*⁻/⁻ mice. D1–D8: *Chst14*⁺/⁺ mice, n = 16; and *Chst14*⁻/⁻ mice, n = 14; D9–D15: *Chst14*⁺/⁺ mice, n = 8; and *Chst14*⁻/⁻ mice, n = 7. (D) Typical colon appearance in *Chst14*⁺/⁺ and *Chst14*⁻/⁻ mice under control conditions, 2.0% DSS (day 8), and 2.0% DSS (day 15). (E) Comparison of colon length (mean ± SEM). (F) Comparison of H&E scores to examine colon injury (mean ± SEM). (A, E, F) *Chst14*⁺/⁺ mice, n = 8;

and *Chst14⁻ᐟ⁻* mice, n = 7. Data were analyzed using one-way ANOVA followed by Tukey–Kramer post hoc test. (E) *P < 0.05, ***P < 0.001, ****P < 0.0001 compared with *Chst14⁺ᐟ⁺* control group. †P < 0.05, ††P < 0.01 compared with 2.0% DSS (day 8) *Chst14⁺ᐟ⁺* group. ‡‡P < 0.01 compared with 2.0% DSS (day 8) *Chst14⁻ᐟ⁻* group. (F) **P < 0.01, ****P < 0.0001 compared with *Chst14⁺ᐟ⁺*control group. ##P < 0.01, ####P < 0.0001 compared with *Chst14⁻ᐟ⁻* control group. †P < 0.05 compared with 2.0% DSS (day 8) *Chst14⁺ᐟ⁺* group. §P < 0.05 compared with 2.0% DSS (day 15) *Chst14⁺ᐟ⁺* group. (G) Colon samples stained with H&E, Alcian blue, and Sirius red under control, 2.0% DSS (day 8), and 2.0% DSS (day 15) conditions. Scale bar: 200 µm. TEM observation of control and 2.0% DSS (day 8 and 15) groups. Scale bar: 500 nm. Cont., control; D8, day 8; D15, day 15; DAI, disease activity index; *Chst14⁺ᐟ⁺* mice, +/+; *Chst14⁻ᐟ⁻* mice, -/-.

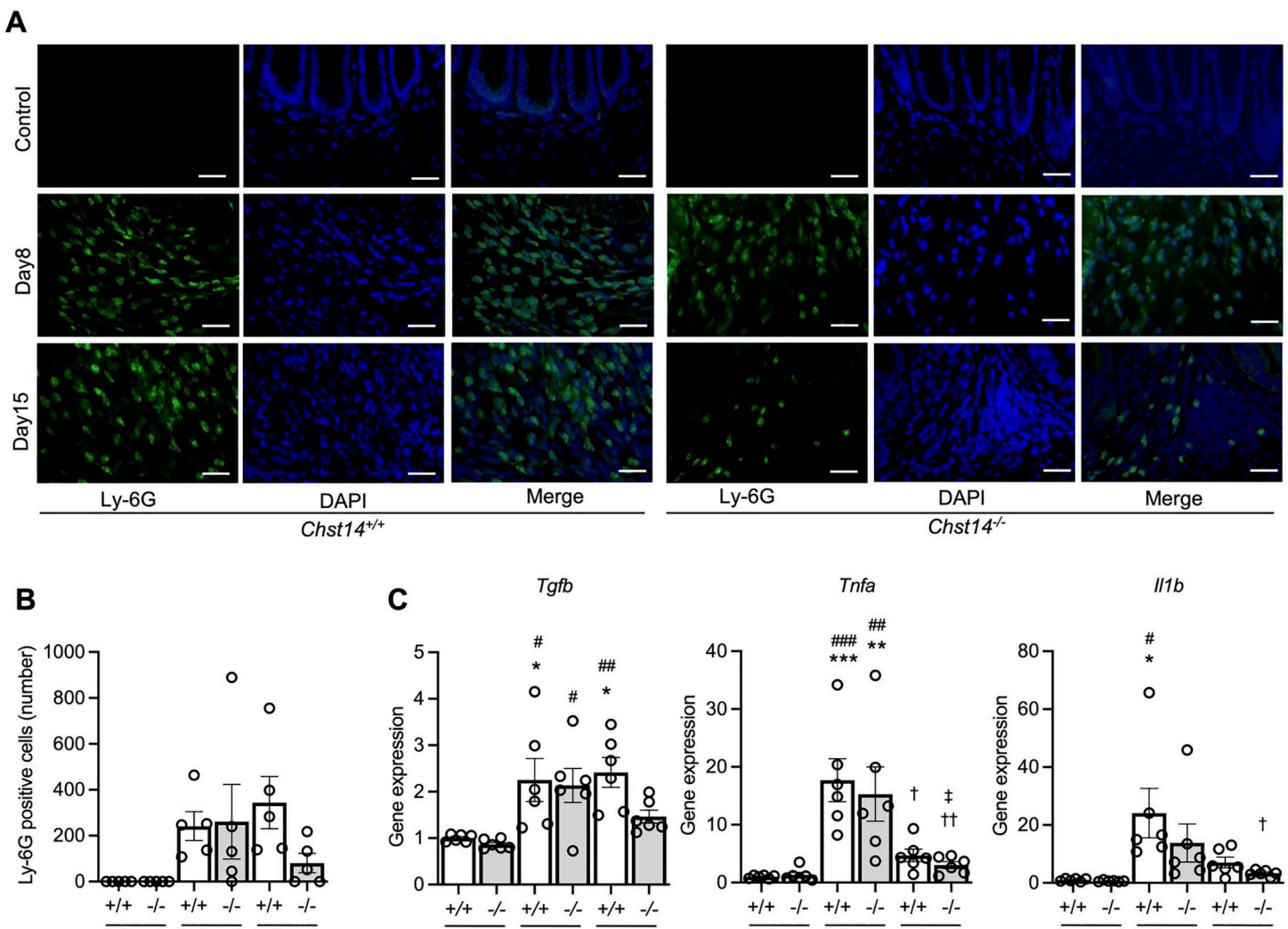

**Fig 5. Fluorescent immunostaining of Ly-6G and gene expression of inflammation-related factors.** (A) Representative results of fluorescent immunostaining of Ly-6G, DAPI, and merged images. (B) Comparison of the number of Ly-6G-positive cells. Values for each individual point reflect the average of three measurements from all fields of a section. Control group – *Chst14⁺ᐟ⁺* and *Chst14⁻ᐟ⁻* mice, n = 5 per group; 2.0% DSS (day 8) group – *Chst14⁺ᐟ⁺* and *Chst14⁻ᐟ⁻* mice, n = 5 per group; 2.0% DSS (day 15) group – *Chst14⁺ᐟ⁺* mice, n = 6; and *Chst14⁻ᐟ⁻* mice, n = 5. Mean ± SEM. (C) Relative mRNA levels of *Tnfa*, *Il1b*, and *Tgfb* in the colon of DSS-induced colitis model mice. *Chst14⁺ᐟ⁺* and *Chst14⁻ᐟ⁻* mice, n = 6 per group. Mean ± SEM. Data were analyzed using one-way ANOVA followed by Tukey–Kramer post hoc test. *P < 0.05, **P < 0.01, ***P < 0.001 compared with control *Chst14⁺ᐟ⁺* group. #P < 0.05, ##P < 0.01, ###P < 0.001 compared with control *Chst14⁻ᐟ⁻* group. †P < 0.05, ††P < 0.01 compared with 2.0% DSS (day 8) *Chst14⁺ᐟ⁺* group. ‡P < 0.05 compared with 2.0% DSS (day 8) *Chst14⁻ᐟ⁻* group. Cont., control; D8, day 8; D15, day 15; *Chst14⁺ᐟ⁺* mice, +/+; *Chst14⁻ᐟ⁻* mice, -/-.

diameters in the submucosa of *Chst14⁻/⁻* mouse colon. Collagen fibrils in the colon are composed of a hybrid of type 1 and 3 collagens [39], with collagen diameter affected by the proportion of type 3 collagen [40]. However, we found no differences in gene expression of molecules related to collagen fibrils (*Col1a1* and *Col3a1*) or core proteins of proteoglycans (*Dcn*, *Bgn*, and *Vcan*) between *Chst14⁻/⁻* and *Chst14⁺/⁺* mice. These results suggest that morphological changes in collagen fibrils in the colon of *Chst14⁻/⁻* mice are not caused by quantitative changes in either collagen molecules or the core proteins of CS/DS-PGs. Interactions between CS/DS-PGs and collagen are reportedly involved in collagen fibril assembly [24]. In the skin of healthy individuals and *Chst14⁺/⁺* mice, CS/DS GAG chains are curved and in close contact along the contours of attached collagen fibrils, whereas in patients with mcEDS-*CHST14* and *Chst14⁻/⁻* mice, the CS-only GAG chains are linear [22]. Biochemical alterations in GAGs resulting from the lack of DS lead to defective assembly of collagen fibrils in the skin [22]. DCN is known as a core protein of CS/DS-PG that binds to type 1 collagen [41]. In this study, DCN was identified as a core protein of CS/DS in the colon. Furthermore, DS on DCN in the colon was significantly reduced and replaced with CS in the absence of *Chst14*. Therefore, it was suggested that the mechanism of defective collagen fibril assembly in the colon is similar to that in the skin.

Furthermore, deformation of collagen fibril assembly is regarded as a cause of skin fragility in patients with mcEDS and *Chst14⁻/⁻* mice [22,24]. In the colon, collagen fibrils are localized in the submucosa and are thought to provide structural strength [42]. In the present study, we found no differences in tissue strength or extensibility, and no signs of tissue fragility in the colon. Unlike in the skin, colon tissue contains muscles that run in multiple directions [43]. Therefore, a limitation of this study is that it may have only evaluated the strength and extensibility of the colon in one direction. The muscular layer of the colon may compensate for tissue fragility caused by structural changes in submucosal collagen fibrils.

Colons were shorter in *Chst14⁻/⁻* mice compared with *Chst14⁺/⁺* mice. We investigated reasons for this shortening of the colon. Administration of the calcium channel blocker, nifedipine (which is also a smooth muscle relaxant), eliminated the difference in colon length between *Chst14⁻/⁻* and *Chst14⁺/⁺* mice. This suggests that colon length in *Chst14⁻/⁻* mice is influenced by intestinal contraction. Intestinal contraction is regulated by smooth muscle, nerves, ion channels, and cell adhesion. However, no differences were detected in the gene expression of the factors related to colonic smooth muscle contraction (S1 Fig A). There is a possibility that the structural changes in collagen fibril assembly influence intestinal contraction, though further research is necessary. We also hypothesized that DS might affect the physiological function of the colon in *Chst14⁻/⁻* mice. However, *Chst14⁻/⁻* mice showed no differences in food intake, defecation volume, gastro-intestinal transit time, and rectoanal reflex. No differences in peristalsis were detected by abdominal ultrasound between *Chst14⁺/⁺* and *Chst14⁻/⁻* mice (S1, S2 Movies, S4 Fig A, B). These data indicate that there are no significant differences in the physiological function of the colon in these mice. Unlike patients with mcEDS-*CHST14*, constipation was not detected in *Chst14⁻/⁻* mice. This may be influenced by the fact that mice do not withhold defecation and lack a sigmoid colon to store stool [43]. Additionally, based on macroscopic and pathological analyses of the colon, we found no diverticula in *Chst14⁻/⁻* mice. Diverticula are believed to develop when the colon tissue weakens and intraluminal pressure increases [44,45], causing the mucosa and submucosa to protrude through the muscularis propria, forming sac-like structures [46]. Diverticula are frequently observed along the tenia coli, which are vulnerable sites for diverticula formation in humans [46,47]. However, mice do not have this structure [48], which along with the absence of these corresponding risk factors, may partially explain the reduced likelihood of diverticula formation.

Age-dependent increases in body weight and colon length were observed in middle-aged *Chst14⁺/⁺* and *Chst14⁻/⁻* mice compared with young mice (S2 Fig A–C). The gene expression of *Col1a1* was decreased in middle-aged *Chst14⁺/⁺* and *Chst14⁻/⁻* mice (S2 Fig J). Similar to young mice, middle-aged *Chst14⁻/⁻* mice also exhibited colon shortening compared with age-matched *Chst14⁺/⁺* mice (S2 Fig A, C). However, no age-related phenotypic changes were observed in the colons of middle-aged *Chst14⁻/⁻* mice (S2 Fig D–I).

Female *Chst14⁻/⁻* mice also showed reduced body weight and colon shortening, similar to male mice (S3 Fig A–C). Furthermore, nifedipine suppressed intestinal contractions and eliminated the difference in colon length between

*Chst14*<sup>+/+</sup> and *Chst14*<sup>-/-</sup> mice (S3 Fig D). On the other hand, gene expression of *Dcn*, collagen-related factor, was significantly decreased in female *Chst14*<sup>-/-</sup> mice compared with that in male *Chst14*<sup>+/+</sup> and *Chst14*<sup>-/-</sup> mice (S3 Fig F). Detailed examination of the physiological functions of the colon in female *Chst14*<sup>-/-</sup> mice is a subject for future research.

Some patients with mcEDS-*CHST14* experience diverticulitis and secondary diverticular perforation [16]. Constipation and increased intestinal pressure are contributing factors to developing diverticulitis, while the mechanisms of inflammation and healing associated with infection and ischemia play critical roles in its progression [49,50]. DS has been reported to interact with growth factors and is associated with wound healing *in vitro* [51]. We next hypothesized that inflammation would be more severe in *Chst14*<sup>-/-</sup> mice. As *Chst14*<sup>-/-</sup> mice did not develop diverticula, the primary cause of diverticulitis, we used the DSS-induced colitis model, a widely recognized model for inflammatory bowel disease, to examine whether DS deficiency influences the severity of inflammation or the healing process [36]. However, under the same dose of DSS intake, there was no difference in the progression of colitis between *Chst14*<sup>-/-</sup> and *Chst14*<sup>+/+</sup> mice. Colon shortening is one of the findings associated with inflammation and is observed in patients with ulcerative colitis and in the DSS colitis model [28]. Significant colon shortening was observed in *Chst14*<sup>+/+</sup> mice but not *Chst14*<sup>-/-</sup> mice. Tissue damage was examined by several histological analyses such as H&E, Alcian blue, and Sirius red staining, and TEM. No analyses revealed evidence of more severe inflammation in *Chst14*<sup>-/-</sup> mice compared with *Chst14*<sup>+/+</sup> mice. Indeed, H&E score suggested that *Chst14*<sup>-/-</sup> mice showed faster histological repair of DSS-induced colitis damage compared with *Chst14*<sup>+/+</sup> mice. Additionally, Ly-6G immunostaining was performed to examine neutrophil infiltration, but no significant differences were observed. Furthermore, gene expression of inflammatory factors such as *Tgfb*, *Tnfa*, and *Il1b*, showed no significant differences between *Chst14*<sup>-/-</sup> and *Chst14*<sup>+/+</sup> mice [52–54]. These results suggest that DS deficiency does not aggravate inflammation and healing in the mouse colon.

## Conclusions

Functions of DS in the colon have not been previously elucidated. In this study, we found that DS deficiency affects collagen assembly and intestinal contraction. However, unlike humans, no significant alterations in physiological function of the colon were observed in mouse. While further detailed investigations are required to fully understand the effects of DS on gastrointestinal physiological functions, this study sheds light on the role of DS in the colon and the potential mechanisms underlying gastrointestinal symptoms in patients with mcEDS.

## Supporting information

**S1 Fig. Gene expression levels related to colonic smooth muscle contraction.** (A) Relative mRNA levels (mean ± SEM) of *Acta2*, *Myh11*, *Mylk*, *Rock1*, *Ryr2*, *Kcnq1*, *Cacna1c*, *Cnn1*, *Calm1*, *Ano1*, *Adrb1*, *Adrb2*, *Chrm2*, *Chrm3*, *Chat*, *Htr1a*, *Htr2a*, *Htr2b*, *Htr3a*, *Htr3b*, *Htr4*, *Htr7*, *Cx26*, and *Gja1* in colon samples from 10-week-old *Chst14*<sup>+/+</sup> and *Chst14*<sup>-/-</sup> mice (n = 6 per group). Gene abbreviations: *Acta2*, actin alpha 2; *Myh11*, myosin heavy chain 11; *Mylk*, myosin light chain kinase; *Rock1*, Rho-associated cooled containing protein kinase 1; *Ryr2*, ryanodine receptor 2; *Kcnq1*, potassium voltage-gated channel subfamily Q member 1; *Cacna1c*, calcium channel voltage-dependent L type alpha 1C subunit; *Cnn1*, calponin 1; *Calm1*, calmodulin 1; *Ano1*, calcium activated chloride channel; *Adrb1*, adrenergic receptor, beta 1; *Adrb2*, adrenergic receptor, beta 2; *Chrm2*, cholinergic receptor, muscarinic 2; *Chrm3*, cholinergic receptor, muscarinic 3; *Chat*, choline acetyltransferase; *Htr1a*, 5-hydroxytryptamine receptor 1A; *Htr2a*, 5-hydroxytryptamine receptor 2A; *Htr2b*, 5-hydroxytryptamine receptor 2B; *Htr3a*, 5-hydroxytryptamine receptor 3A; *Htr3b*, 5-hydroxytryptamine receptor 3B; *Htr4*, 5-hydroxytryptamine receptor 4; *Htr7*, 5-hydroxytryptamine receptor 7; *Cx26*, connexin-26; *Gja1*, gap junction protein alpha 1.
(PDF)

**S2 Fig. Colonic morphology, function, and collagen-related gene expression in middle-aged mice.** (A) Typical appearance of the colon in middle-aged mice. (B) Body weight (mean±SEM). (C) Comparison of colon length between young and middle-aged groups (mean±SEM). (B), (C) Young *Chst14*[+/+] mice, n=7; and *Chst14*[-/-] mice, n=7; and middle-aged *Chst14*[+/+] mice, n=8; and *Chst14*[-/-] mice, n=9. Data were analyzed using one-way ANOVA followed by Tukey–Kramer post hoc test. *P<0.05, **P<0.01, ***P<0.001 compared with young *Chst14*[+/+] group. ##P<0.01, ####P<0.0001 compared with young the *Chst14*[-/-] group. †P<0.05, ††††P<0.0001 compared with middle-aged *Chst14*[+/+] group. (D) H&E-stained colon sections from middle-aged *Chst14*[+/+] and *Chst14*[-/-] mice. Scale bar: 200 μm. (E) Amount of defecation per day. (F) Daily food intake. (E, F) Young *Chst14*[+/+] mice, n=8; and young *Chst14*[-/-] mice, n=8; and middle-aged *Chst14*[+/+] mice, n=8; and middle-aged *Chst14*[-/-] mice, n=9. (G) Load required for colon rupture. (H) Rate of colon extension when loaded with a 10 g weight. (G, H) Young *Chst14* [+/+] mice, n=12; and young *Chst14*[-/-] mice, n=6; and middle-aged *Chst14*[+/+] mice, n=8; and middle-aged *Chst14*[-/-] mice, n=9. (I) Gastrointestinal transit time. Young *Chst14*[+/+] mice, n=9; and young *Chst14*[-/-] mice, n=8; and middle-aged *Chst14*[+/+] mice, n=8; and middle-aged *Chst14*[-/-] mice, n=9. (J) Relative mRNA levels of *Col1a1*, *Col3a1*, *Dcn*, *Bgn*, and *Vcan* in the colon (mean±SEM) analyzed by one-way ANOVA followed by Tukey–Kramer post hoc test. Young and middle-aged *Chst14*[+/+] and *Chst14*[-/-] mice (n=6 per group). Young mice, 10-week-old male mice; middle-aged mice, 36- to 40-week-old male mice. *Chst14*[+/+] mice,+/+; *Chst14*[-/-] mice, -/-.
(PDF)

**S3 Fig. Colonic morphology and collagen-related gene expression in female mice.** (A) Typical appearance of the colon in female mice. (B) Comparison of body weight between female *Chst14*[+/+] and *Chst14*[-/-] mice. (mean±SEM). ***P<0.001 compared with *Chst14*[+/+]. (C) Comparison of colon length between female *Chst14*[+/+] and *Chst14*[-/-] mice. (mean±SEM). *P<0.05 compared with *Chst14*[+/+]. (B, C) *Chst14*[+/+] mice, n=5 and *Chst14*[-/-] mice, n=3. Data were analyzed using *t*-test. (D) Comparison of colon length between pre- and post-nifedipine groups (mean±SEM). Pre- and post-nifedipine *Chst14*[+/+] mice (n=5) and *Chst14*[-/-] mice (n=3). Data were analyzed using one-way ANOVA followed by Tukey–Kramer post hoc test. *P<0.05 compared with pre-nifedipine *Chst14*[+/+] group. ##P<0.001 compared with pre-nifedipine *Chst14*[-/-] group. (E) H&E-stained colon sections from *Chst14*[+/+] and *Chst14*[-/-] mice. Scale bar: 200 μm. (F) Relative mRNA levels of *Col1a1*, *Col3a1*, *Dcn*, *Bgn*, and *Vcan* in the colon (mean±SEM) analyzed by one-way ANOVA followed by Tukey–Kramer post hoc test. Male and female *Chst14*[+/+] and *Chst14*[-/-] mice (n=6 per group). *P<0.05 compared with male *Chst14*[+/+] group. #P<0.05 compared with male *Chst14*[-/-] group. *Chst14*[+/+] mice,+/+; *Chst14*[-/-] mice, -/-; Male, 10-week-old male mice; female, 12-week-old female mice.
(PDF)

**S4 Fig. Still images extracted from ultrasound videos showing the colons of *Chst14*[+/+] and *Chst14*[-/-] mice.**
(PDF)

**S1 Movie. A representative ultrasound video of the colon from a *Chst14*[+/+] mouse.** Abdominal ultrasound was performed for more than 10 minutes on each of the five male *Chst14*[+/+] mice (11 weeks old), and a representative 8-second video was obtained.
(MP4)

**S2 Movie. A representative ultrasound video of the colon from a *Chst14*[-/-] mouse.** Abdominal ultrasound was performed for more than 10 minutes on each of the three male *Chst14*[-/-] mice (11–12 weeks old), and a representative 8-second video was obtained.
(MP4)

**S1 Table. RT-qPCR primer sequences.**
(PDF)

**S2 Table. Relative mRNA levels.**
(PDF)

**S3 Table. Disaccharide composition of CS/DS, CS, and DS.**
(PDF)

**S4 Table. Physiological measurements.**
(PDF)

**S5 Table. Measured thicknesses of each layer.**
(PDF)

**S6 Table. Measured diameter of collagen fibrils.**
(PDF)

**S7 Table. Data of DSS colitis model.**
(PDF)

**S8 Table. Number of Ly-6G-positive cells.**
(PDF)

**S1 File. Raw images.** Uncropped Western blot images for DCN and GAPDH.
(PDF)

## Acknowledgments

We are grateful to the Division of Instrumental Research, Research Center for Advanced Science and Technology, Shinshu University, for its facilities and scientific and technical support. We thank all the staff of the Division of Animal Research, Research Center for Advanced Science and Technology, Shinshu University for their cooperation in the experiments. The authors thank Alison McTavish, MSc from Edanz (https://jp.edanz.com/ac) for editing a draft of this manuscript.

## Author contributions

**Conceptualization:** Fumiko Ono, Takahiro Yoshizawa.

**Data curation:** Fumiko Ono, Takahiro Yoshizawa.

**Formal analysis:** Fumiko Ono, Shinji Miyata, Takahiro Yoshizawa.

**Funding acquisition:** Tomoki Kosho, Takahiro Yoshizawa.

**Investigation:** Fumiko Ono.

**Methodology:** Fumiko Ono, Yuki Takahashi, Shuji Mizumoto, Shinji Miyata, Shuhei Yamada, Takahiro Yoshizawa.

**Project administration:** Tomoki Kosho, Takahiro Yoshizawa.

**Resources:** Fumiko Ono, Yuki Takahashi, Shin Shimada, Yuko Nitahara-Kasahara, Takashi Okada, Takahiro Yoshizawa.

**Supervision:** Tomoki Kosho, Takahiro Yoshizawa.

**Validation:** Fumiko Ono, Takahiro Yoshizawa.

**Visualization:** Fumiko Ono, Takahiro Yoshizawa.

**Writing – original draft:** Fumiko Ono, Shinji Miyata, Takahiro Yoshizawa.

**Writing – review & editing:** Yuki Takahashi, Shuji Mizumoto, Shuhei Yamada, Tomoki Kosho, Takahiro Yoshizawa.

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
