## [Decision Letter · Decision Letter 0]

3 Jan 2025

PONE-D-24-57016Carbohydrate sulfotransferase 14 gene deletion induces dermatan sulfate deficiency and affects collagen structure and bowel contractionPLOS ONE

Dear Dr. Yoshizawa,

Thank you for submitting your manuscript to PLOS ONE. After careful consideration, we feel that it has merit but does not fully meet PLOS ONE’s publication criteria as it currently stands. Therefore, we invite you to submit a revised version of the manuscript that addresses the points raised during the review process.

We look forward to receiving your revised manuscript.

Kind regards,

Nikos K Karamanos, Ph.D.

Academic Editor

PLOS ONE

Journal Requirements:

2. To comply with PLOS ONE submissions requirements, in your Methods section, please provide additional information regarding the experiments involving animals and ensure you have included details on (1) methods of sacrifice, (2)and  efforts to alleviate suffering.

Additional Editor Comments:

It is an interesting and well contacted experiment study. Points given by rev I are valid and well could improve the quality of this study further. Therefore authors may revise the manuscript accordingly. It may be useful that authors give recent refs in the theme,

Reviewers' comments:

Reviewer's Responses to Questions

**Comments to the Author**

1. Is the manuscript technically sound, and do the data support the conclusions?

Reviewer #1: Partly

2. Has the statistical analysis been performed appropriately and rigorously? 

Reviewer #1: Yes

3. Have the authors made all data underlying the findings in their manuscript fully available?

Reviewer #1: Yes

4. Is the manuscript presented in an intelligible fashion and written in standard English?

Reviewer #1: Yes

5. Review Comments to the Author

Reviewer #1: The work by Fumiko Ono aimed To determine the effects of DS deficiency on the gastrointestinal tract. The experimental procedures outlined in the manuscript provide a comprehensive approach to investigating the effects of dermatan sulfate (DS) deficiency on colonic structure and function in Chst14 knockout mice. The authors performed a variety of experiments, including histological analyses, collagen fibril assessments, and a dextran sulfate sodium (DSS) colitis model, to evaluate the physiological and pathological consequences of DS deficiency.

Support for Conclusions:

- Collagen Structure: The results indicate that collagen fibrils in the colon of Chst14-/- mice were abnormally arranged and exhibited smaller diameters compared to wild-type mice, which supports the conclusion that DS deficiency affects collagen assembly.

- Colonic Function: The findings of increased colonic contraction and shorter colon length in Chst14-/- mice suggest that DS plays a role in regulating colonic physiology. However, the lack of significant differences in defecation volume, food intake, and gastrointestinal transit time implies that physiological function may not be severely impacted in the same way it is in humans with mcEDS.

- DSS Colitis Model: The study found no significant aggravation of colitis in Chst14-/- mice compared to wild-type mice, suggesting that DS deficiency does not worsen inflammation or healing in the colonic context.

Critiques:

- In-depth Mechanistic Studies: The study could benefit from further mechanistic investigations to elucidate the specific pathways through which DS affects collagen assembly and colonic contraction. This could include exploring the molecular interactions between DS and collagen or other extracellular matrix components.

- Broader Physiological Assessments: Additional experiments assessing other aspects of gastrointestinal function, such as motility patterns or responses to food intake, could provide a more complete picture of how DS deficiency affects the colon.

- Longitudinal Studies: Conducting longitudinal studies could help determine if the observed effects in Chst14-/- mice change over time or with age, which may provide insights into the progression of symptoms that occur in patients with mcEDS.

- Inclusion of Female Mice: The study primarily used male mice to avoid gender-related variations. Including female mice in future experiments could provide more comprehensive data and help understand any potential gender differences in DS function.

- Comparative Studies with Human Tissue: Comparing the findings in mouse models with human tissue samples from mcEDS patients could validate the translational relevance of the animal model and refine the conclusions drawn from the study.

Overall, while the experimental procedures and results support the conclusions, the study could be strengthened by addressing the critiques and incorporating the suggestions mentioned above to provide a deeper understanding of DS's role in colonic physiology and pathology.

6. PLOS authors have the option to publish the peer review history of their article (what does this mean? ). If published, this will include your full peer review and any attached files.

**Do you want your identity to be public for this peer review?** For information about this choice, including consent withdrawal, please see our Privacy Policy .

Reviewer #1: **Yes: ** Mauro Sergio Pavao

---

## [Author Response · Author response to Decision Letter 1]

24 Feb 2025

Please find attached file, "Response to Reviewers".

---

## [Editor Report · Decision Letter 1]

27 Feb 2025

Carbohydrate sulfotransferase 14 gene deletion induces dermatan sulfate deficiency and affects collagen structure and bowel contraction

PONE-D-24-57016R1

Dear Dr. Yoshizawa,

We’re pleased to inform you that your manuscript has been judged scientifically suitable for publication and will be formally accepted for publication once it meets all outstanding technical requirements.

Kind regards,

Nikos K Karamanos, Ph.D.

Academic Editor

PLOS ONE
---

## [Editor Report · Acceptance letter]

PONE-D-24-57016R1

PLOS ONE

Dear Dr. Yoshizawa,

I'm pleased to inform you that your manuscript has been deemed suitable for publication in PLOS ONE. Congratulations! Your manuscript is now being handed over to our production team.

Kind regards,

on behalf of

Prof. Dr. Nikos K Karamanos

Academic Editor

PLOS ONE